# Impact of Phyllosphere *Methylobacterium* on Host Rice Landraces

Pratibha Sanjenbam,[a] P. V. Shivaprasad,[a] Deepa Agashe[a]

[a]National Center for Biological Sciences, Tata Institute of Fundamental Research, Bangalore, India

**ABSTRACT** The genus *Methylobacterium* includes widespread plant-associated bacteria that are abundant in the plant phyllosphere (leaf surfaces), consume plant-secreted methanol, and can produce plant growth-promoting metabolites. However, despite the potential to increase agricultural productivity, their impact on host fitness in the natural environment is relatively poorly understood. Here, we conducted field experiments with three traditionally cultivated rice landraces from northeastern India. We inoculated seedlings with native versus nonnative phyllosphere *Methylobacterium* strains and found significant impacts on plant growth and grain yield. However, these effects were variable. Whereas some *Methylobacterium* isolates were beneficial for their host, others had no impact or were no more beneficial than the bacterial growth medium on its own. Host plant benefits were not consistently associated with *Methylobacterium* colonization and did not have altered phyllosphere microbiome composition, changes in the early expression of plant stress response pathways, or bacterial auxin production. We provide the first demonstration of the benefits of phyllosphere *Methylobacterium* for rice yield under field conditions and highlight the need for further analysis to understand the mechanisms underlying these benefits. Given that the host landrace-*Methylobacterium* relationship was not generalizable, future agricultural applications will require careful testing to identify coevolved host-bacterium pairs that may enhance the productivity of high-value rice varieties.

**IMPORTANCE** Plants are associated with diverse microbes in nature. Do the microbes increase host plant health, and can they be used for agricultural applications? This is an important question that must be answered in the field rather than in the laboratory or greenhouse. We tested the effects of native, leaf-inhabiting bacteria (genus *Methylobacterium*) on traditionally cultivated rice varieties in a crop field. We found that inoculation with some bacteria increased rice grain production substantially while a nonnative bacterium reduced plant health. Overall, the effect of bacterial inoculation varied across pairs of rice varieties and their native bacteria. Thus, knowledge of evolved associations between specific bacteria hosted by specific rice varieties is necessary to develop ways to increase the yield of traditional rice landraces and preserve these important sources of cultural and genetic diversity.

**KEYWORDS** host-bacterial interaction, host fitness, phyllosphere, epiphytes, grain yield, rice landraces

Address correspondence to Deepa Agashe, dagashe@ncbs.res.in.

[This article was published on 20 July 2022 with the incorrect affiliation. The affiliation has been updated in the current version, posted on 27 July 2022.]

The authors declare no conflict of interest.

Plants are associated with a diverse set of microbes, both belowground ("rhizosphere") and aboveground ("phyllosphere"). The phyllosphere harbors nearly $10^7$ bacterial cells/$cm^2$ of leaf surface with both biotic and abiotic factors influencing the composition of bacterial communities (1, 2). The importance of phyllosphere microbes is highlighted by their large-scale impacts. For instance, leaf microbiome diversity contributes significantly to the productivity of tree communities (3). Hence, a large body of work has focused on understanding the establishment and stability of phyllosphere communities as well as the interactions between community members and host plants. For example, in both wild and domesticated plants, microbial communities tend to be hierarchically structured. Soil

microbiomes harbor maximum diversity and complexity followed by rhizosphere and phyllosphere epiphytes that colonize leaf and stem surfaces, and finally, phyllosphere endophytes that colonize internal leaf tissues (4–8). An important factor driving microbial assembly in the phyllosphere is the strength and nature of selection acting at the leaf surface. For instance, phyllosphere bacteria must deal with both antimicrobials and limited nutrients in plant leaf exudates and exposure to UV radiation and desiccation (1, 2). These factors may impose selection for bacterial traits such as aggregation, deployment of efflux pumps, and the production of protective pigments and biosurfactants. Indeed, the composition of phyllosphere bacterial communities, which are typically dominated by Alphaproteobacteria and Gammaproteobacteria (9), suggests a role for selection rather than stochastic community assembly. However, the nature, strength, and variability of such selection remains poorly understood.

Recent work does show broad potentially useful functions provided by phyllosphere bacteria, including nitrogen fixation (4, 10, 11). However, whether and how these functions directly benefit the host plant is not well understood in most cases. Some of the best examples of direct fitness benefits to the host derive from studies on abundant phyllosphere bacteria from the genus *Methylobacterium*. This group includes species that consume methanol released by plant cell walls, such that methanol metabolism is key during phyllosphere colonization (7). Many species from this group are thought to enhance host plant fitness (12, 13). For example, epiphytic *Methylobacterium* inoculation promotes moss seed growth to a level comparable to the effects of applying synthetic cytokinin (14). Similar results are observed with cell-free culture suspensions of *Methylobacterium* growth medium, which increase germination and seedling growth in wheat, potentially due to cytokinins secreted by the bacteria (15). Some *Methylobacterium* strains enhance seed ripening in rice, though they have no impact on barley yield (16). Many *Methylobacterium* strains produce (or have the necessary genes or metabolic pathways for) several plant growth-promoting factors such as cytokinins, that could confer significant fitness benefits on the host plant (12, 13, 17). However, prior studies have largely been conducted in the laboratory or greenhouse, where the natural environmental context is missing. It is, therefore, unclear whether the observed impacts of *Methylobacterium* can be extrapolated to field conditions. Furthermore, many studies use standard plant model systems to test the impact of bacteria isolated from various sources. To fully understand and use plant-microbe relationships, we must evaluate them in the field while maintaining the context of naturally evolved associations.

Here, we tested the specificity of host-microbial relationships in three traditionally cultivated rice landraces from northeast India. In this region, distinct rice landraces have been cultivated locally for many generations and show substantial genotypic and phenotypic divergence (18). Previously, we showed that the host rice landrace was the best predictor of phenotypic variation in epiphytic *Methylobacterium* strains, with different landraces associating with distinct *Methylobacterium* strains (19). Preliminary greenhouse experiments further showed that some rice landraces gained an early growth advantage when inoculated with *Methylobacterium* strains isolated from their leaves. Therefore, we hypothesized that *Methylobacterium* strains may confer a fitness advantage to their specific host plants during rice cultivation. To test this, we chose three phenotypically distinct landraces from our previous study that are cultivated in the state of Manipur. We inoculated rice seedlings with a native *Methylobacterium* strain isolated from the specific landrace ("own" bacteria), or a nonnative strain from a different landrace ("other" bacteria) and measured plant growth and yield-related traits in an experimental field plot in Manipur (Fig. 1). Local adaptation between bacteria and hosts might result in host-specific fitness benefits, with potential implications for increasing agricultural productivity of the landraces. We found that the impacts of phyllosphere *Methylobacterium* varied across landraces. While some *Methylobacterium* strains were beneficial for their host plant, others either had no impact or were as beneficial as the bacterial growth medium control. In contrast, inoculating plants with the nonnative *Methylobacterium* strain generally reduced plant fitness. Our results suggest

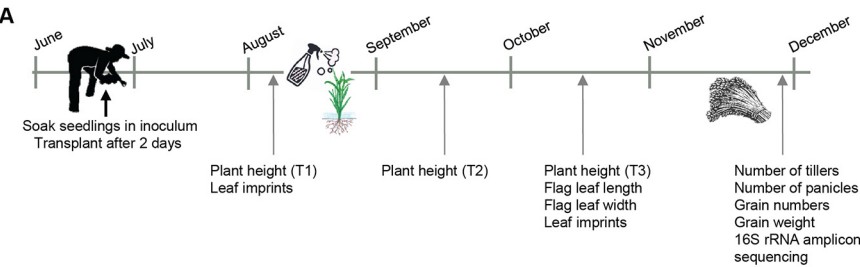

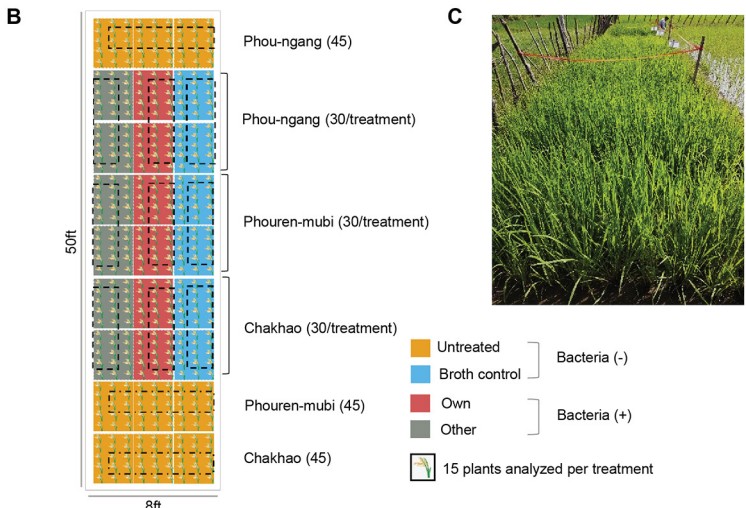

**FIG 1** Experimental design. (A) Timeline of the field experiment with key events. (B) Schematic showing the field design. The total number of plants transplanted is given in parentheses. (C) A view of the field site 1 month after transplantation.

that understanding locally evolved plant-microbial relationships in the phyllosphere is critical for developing agricultural applications.

## RESULTS

To test the impact of *Methylobacterium* on host rice plants, we designed field experiments incorporating four treatments in three rice landraces (Chakhao, Phouren-mubi, and Phou-ngang) from Manipur (Fig. 1). "Untreated" seedlings were exposed to water before transplanting in soil and during foliar spraying; "broth control" seedlings were exposed to sterile bacterial growth medium; "own" seedlings were exposed to a growing culture of native *Methylobacterium* that was originally isolated from the flag leaves of the specific landrace; "other" seedlings were exposed to a *Methylobacterium* strain that was originally isolated from a landrace from Arunachal Pradesh (Fig. 1). Based on potential interactions between the rice landraces and *Methylobacterium* strains, we outlined several possible outcomes and implications of the field experiment.

(i) *Methylobacterium* may not increase host fitness relative to the broth control in any landrace (Fig. 2A).

(ii) *Methylobacterium* may only be beneficial to its "own" landrace. In this case, we expected the fitness of "own" plants to be higher than that of "broth control" plants, but the fitness with "other" treatment would be similar to that of "broth" (Fig. 2B).

(iii) Non-native *Methylobacterium* may be harmful to the host. In this case, we expected the fitness of "other" plants to be lower than that of "broth" and "own" plants (Fig. 2C).

(iv) *Methylobacterium* may confer a general fitness advantage that is not specific to their native host landrace. In this case, we expected the fitness of "own" and "other" treatments to be similar, and higher than "broth" (Fig. 2D).

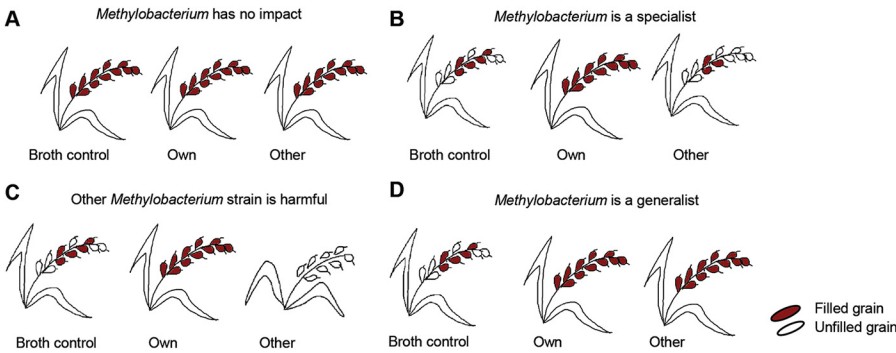

**FIG 2** Illustration of possible outcomes of the field experiment, and their implications. (A) *Methylobacterium* did not provide a host fitness advantage relative to the broth control. (B) *Methylobacterium* was only beneficial for its "own" landrace. (C) Nonnative *Methylobacterium* was harmful to the host. (D) *Methylobacterium* conferred a general fitness advantage that is not specific to the native host landrace.

To test these possible interactions, we measured multiple phenotypes relevant for plant vegetative growth as well as reproduction. We noted that the broth control would have more available nutrients than the other treatments where the bacteria would convert a substantial fraction of nutrients into biomass. Therefore, we would expect that the growth medium alone would have a greater ability to support plant fitness in the broth control. Hence, compared to the broth control, we would expect to obtain a conservative estimate of the impact of bacterial inoculation.

**The impact of *Methylobacterium* strains varied across landraces.** Two important crop plant traits are growth rate and seed yield, which served as indicators of plant fitness. During the growing period, the total increase in plant height (i.e., T3 to T1, Fig. 1A) was broadly similar across all four treatments. However, treatment with broth alone or broth containing *Methylobacterium* reduced growth compared to untreated plants of Chakhao and Phou-ngang (Fig. 3A; Table S3A; Fig. S3). In Phou-ngang, treatment with nonnative ("other") *Methylobacterium* caused a further reduction in growth compared to the other two treatments (Fig. 3A). In contrast, total grain yield (i.e., total grain weight per plant) was typically higher for treated plants of all landraces, with the highest yield observed in both control and/or "own" treatments (Fig. 3B; Table S3A). These patterns were robust to the inclusion of potential outliers (Fig. 3). The overall weak impact of native *Methylobacterium* on vegetative growth was also observed for other growth-related traits (Fig. S4), while additional reproductive traits such as the number and proportion of filled grains increased in at least two landraces, Chakhao and Phouren-mubi (Fig. S5). Thus, *Methylobacterium* tended to increase grain yield but had little impact on vegetative plant growth across landraces.

To better understand these effects and test our specific predictions (Fig. 2), we calculated the difference in median trait values for the following pairs of treatments, for each landrace. (i) Broth control – untreated, which was a measure of the effect of the broth alone; (ii) own – broth control, a measure of the effect of own *Methylobacterium* over and above any broth effects; (iii) other – broth control, reflecting the effect of nonnative *Methylobacterium*. Surprisingly, the broth alone had a significant impact on nearly all plant traits, with a substantial (2 to 4-fold) increase in the number and fraction of filled grains as well as total grain yield in broth-treated plants compared to untreated plants of two out of three landraces (Fig. 4A and B; Table S4). On the other hand, growth rate and plant height showed a significant though small reduction due to broth in two landraces. All other broth effects were observed in only a single landrace. Altogether, broth enhanced some, but not all, aspects of plant fitness. Note that one limitation of our experiment was that we could not implement a randomized block design. However, it was unlikely that systematic environmental gradients explain our results because we did not see the "other" or "broth control" treatments being consistently the best or the worst across landraces (Fig. 1A).

Next, we tested the impact of the specific phyllosphere *Methylobacterium* associated with each rice landrace after accounting for the broth effects described above.

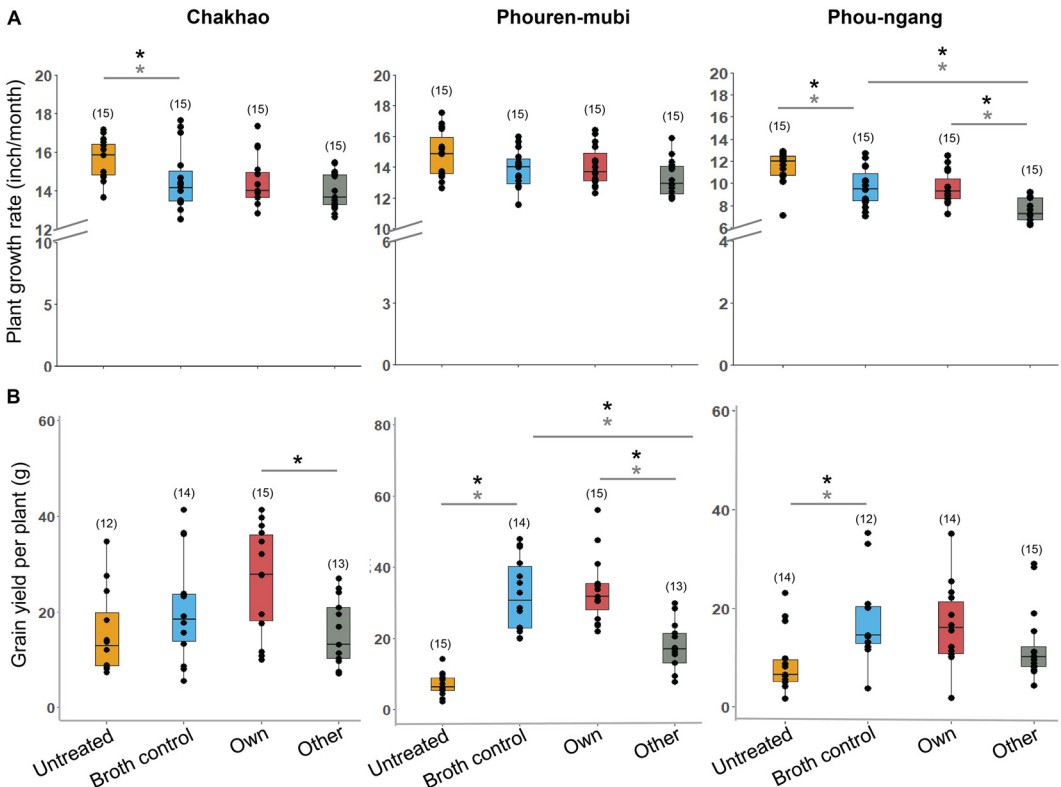

**FIG 3** The impact of *Methylobacterium* inoculation varies across landraces. Boxplots show (A) plant growth rate (change in height) and (B) yield, as a function of different treatments across landraces. Sample size (number of replicate plants) is indicated in parentheses; asterisks indicate significant pairwise differences ($P < 0.05$) when including all replicates (gray) and when excluding influential data points (black), as estimated using GLM/ANOVA followed by Tukey's HSD.

Because the "own" treatment included broth, any differences between the broth control and "own" can be attributed to the bacteria. As observed for the broth control, the magnitude of the impact of "own" *Methylobacterium* also varied across landraces (Fig. 4C; Table S3C), although there were some qualitative similarities in that all significant impacts were positive (Fig. 4D). The number of tillers and percentage of filled grains tended to be higher in all landraces, although this was significant only in one landrace. In Chakhao, *Methylobacterium* exaggerated the effect of broth on grain filling with a 30% increase in the number of filled grains and a 13% increase in grain filling. However, in Phouren-mubi, *Methylobacterium* only increased the number of tillers, and similar to Chakhao, lead to a small increase in the flag leaf length. Finally, in Phou-ngang, we did not observe a significant change in any of the traits. Thus, in contrast to the broth treatment, own *Methylobacterium* had relatively rare, although in some cases very large impacts on their host plants. These results partially supported our second prediction that *Methylobacterium* could be beneficial for their host plants.

Finally, we found that treatment with "other" (nonnative) *Methylobacterium* was consistently deleterious with large reductions in grain filling and yield relative to the broth control (Fig. 4E and F; Table S3D). In many cases, the reduction in trait values in the "other" treatment was similar to or lower than the observed benefit of the broth. For instance, Chakhao plants in the broth control treatment had ∼300 more filled grains per plant compared to untreated plants (Fig. 4A). But plants in the "other" treatment had ∼300 fewer grains than broth control, suggesting that the "other" *Methylobacterium* prevents the plant from gaining the benefits of the broth. Note that the same strain was used for the "other" treatment of all landraces; hence, it is also possible that this strain is peculiar in its inability to provide fitness benefits or imposes a general fitness cost on rice hosts. Further, because this bacterial strain was originally isolated from a rice landrace in Arunachal Pradesh (a different state in the

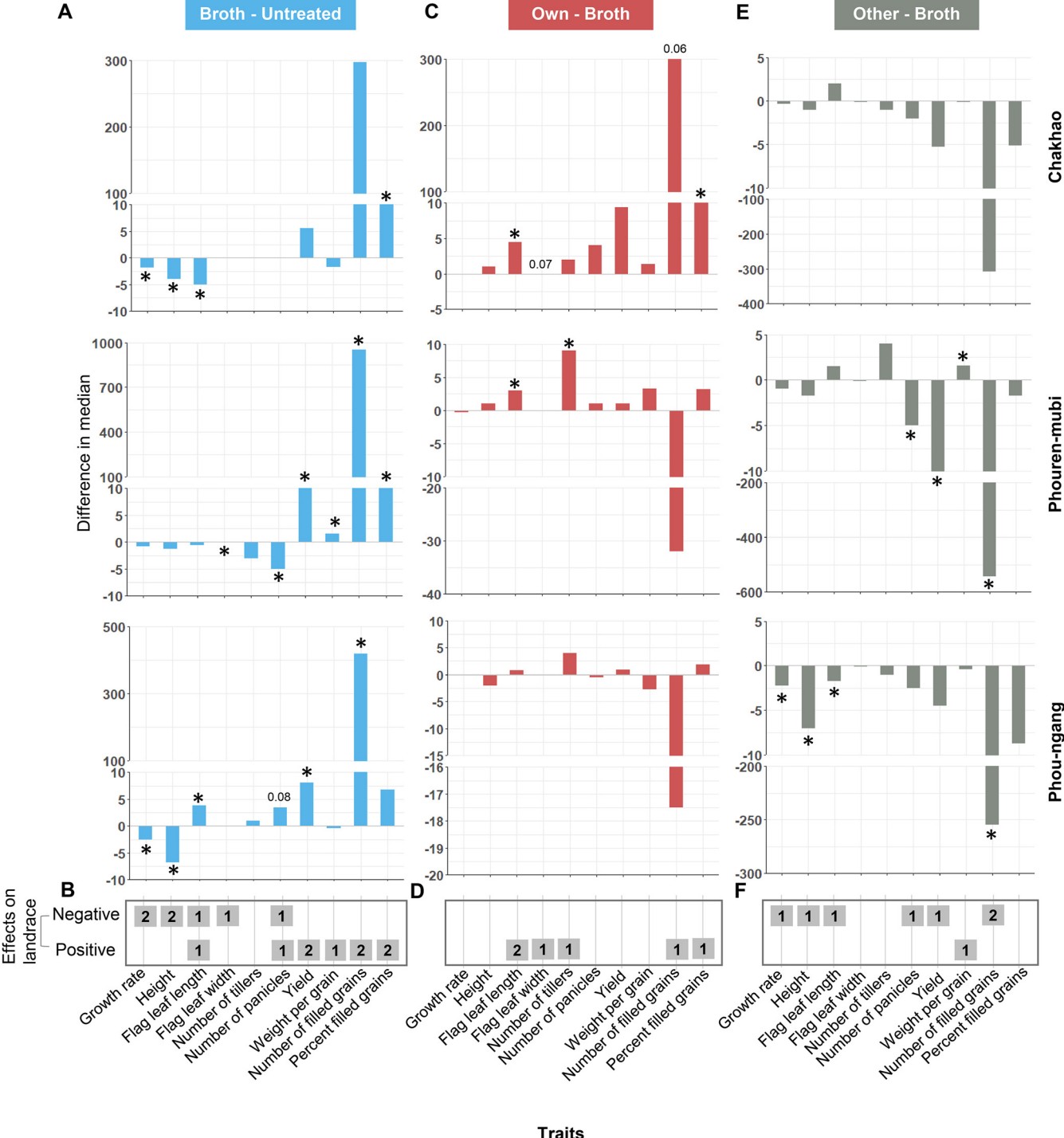

**FIG 4** Summary of the impact of experimental treatments on plant phenotype. Bar plots show the impact of (A and B) broth alone (compared to untreated), (C and D) own *Methylobacterium* (relative to the broth), and (E and F) other *Methylobacterium* (relative to the broth) on all plant traits measured in this study after removing influential data points (Supplemental File 1). The bar height indicates the difference in median trait values across the two relevant treatments. Asterisks indicate values that are significantly different from zero ($P < 0.05$); $P$ values close to 0.05 were noted explicitly (Table S4). (B, D, and F) The number of landraces in which a significant effect was observed (positive or negative); e.g., "2" indicates that two landraces showed a similar impact of the experimental treatment.

region), the landraces used in our experiments may not have encountered this strain in the past, and therefore may not be able to derive fitness benefits from that strain.

Overall, in Chakhao, inoculation with landrace-associated phyllosphere *Methylobacterium* had a large positive impact on yield-related traits but little impact on vegetative growth. In

contrast, in Phouren-mubi, the native *Methylobacterium* strain increased aspects of vegetative growth (tiller number) but tended to reduce grain filling compared to the broth control, although this was not significant. Interestingly, the broth alone had a very large effect on grain filling in this landrace (~1000 more grains per plant compared to untreated), potentially masking any benefits that may be conferred by the bacteria. Finally, in the case of Phou-ngang, we did not find any significant host plant benefits from phyllosphere-associated *Methylobacterium*. Altogether, these results seem to support our third prediction that nonnative *Methylobacterium* were detrimental (see the discussion for caveats) but contradict our fourth hypothesis stating that the benefits of *Methylobacterium* were generalized. Thus, while rice plants did not acquire similar benefits from randomly chosen *Methylobacterium* strains, native bacterial strains were also not guaranteed to be beneficial across landraces.

**The phyllosphere microbiome was unaltered by *Methylobacterium* inoculation.** Several studies indicate that the beneficial effects of some bacteria may involve the exclusion of harmful colonizers or facilitation of other growth-promoting bacteria. To explore the causes of the observed impact on host fitness and yield, we first tested whether the inoculated *Methylobacterium* strains successfully colonized the experimental plants. Unexpectedly, 2 months after the foliar spraying we only found a few *Methylobacterium* colonies in leaf imprints (Fig. S6). Most recovered strains (identified using 16S rRNA sequencing) did not match the experimentally inoculated strains, except in Chakhao where the inoculated "own" strain appeared to have colonized the treated plants (Table S5). Interestingly, the "other" *Methylobacterium* strain was not observed in any landrace, suggesting that this strain is a poor colonizer of these landraces in the environmental conditions prevalent in Manipur. These results also suggested that the effect seen on yield must have been due to the immediate impacts on the host after inoculation.

Next, we tested whether the impact of *Methylobacterium* inoculation on host plant fitness might be correlated with the altered composition of the phyllosphere microbiome. We focused on Chakhao because it showed the strongest fitness benefits of its native *Methylobacterium*. By the time of harvesting, the flag leaf microbiome was not significantly different across treatments (Fig. 5A). Further, we tested whether specific functional groups of bacteria were enriched or depleted across treatments. We used broad taxonomic identification of community members to determine the relative abundance of all bacteria from the genus *Methylobacterium* ("total M"), taxa that were closely related to the experimentally inoculated "own," "other", or environmental *Methylobacterium*, as well as common pathogens of rice plants such as *Burkholderia*, *Pseudomonas*, and *Xanthomonas*. As expected, the genus *Methylobacterium* constituted ~10 to 20% of all bacterial reads of phyllosphere microbiomes in all treatments (Fig. 5B). Surprisingly, the proportion of pathogens was largest in untreated plants (Fig. 5B) and was not correlated with the abundance of all *Methylobacterium* (Spearman's rho = −0.13, $P = 0.6$) or of the inoculated (own) *Methylobacterium* (Spearman's rho = −0.144, $P = 0.63$). Interestingly, the "own" *Methylobacterium* strain constituted 60 to 70% of all *Methylobacterium* across treatments (including plants inoculated with the "other" *Methylobacterium*; Fig. 5C), confirming that this specific strain successfully colonizes Chakhao plants in the field, as observed from leaf imprints. We also observed additional (likely environmentally acquired) *Methylobacterium* strains on all plants, although at a much lower abundance. Very rarely did we observe strains that closely matched the inoculated "other" strain (Fig. 5C). In the other two landraces, the abundance of pathogens was generally lower than in Chakhao, and the inoculated native *Methylobacterium* strains were also not abundant (Fig. S7).

We note some limitations of these data. First, the low sample sizes for some treatments limit our ability to draw strong conclusions about microbiome variation. Because we collected the flag leaf microbiomes during harvesting, we also did not know whether the observed bacterial taxa colonized the flag leaves during or after plant growth or reproduction, impeding our ability to infer causal links. Third, because the 16S rRNA gene offered limited resolution, we could not be certain about accurate identification of the "own" and

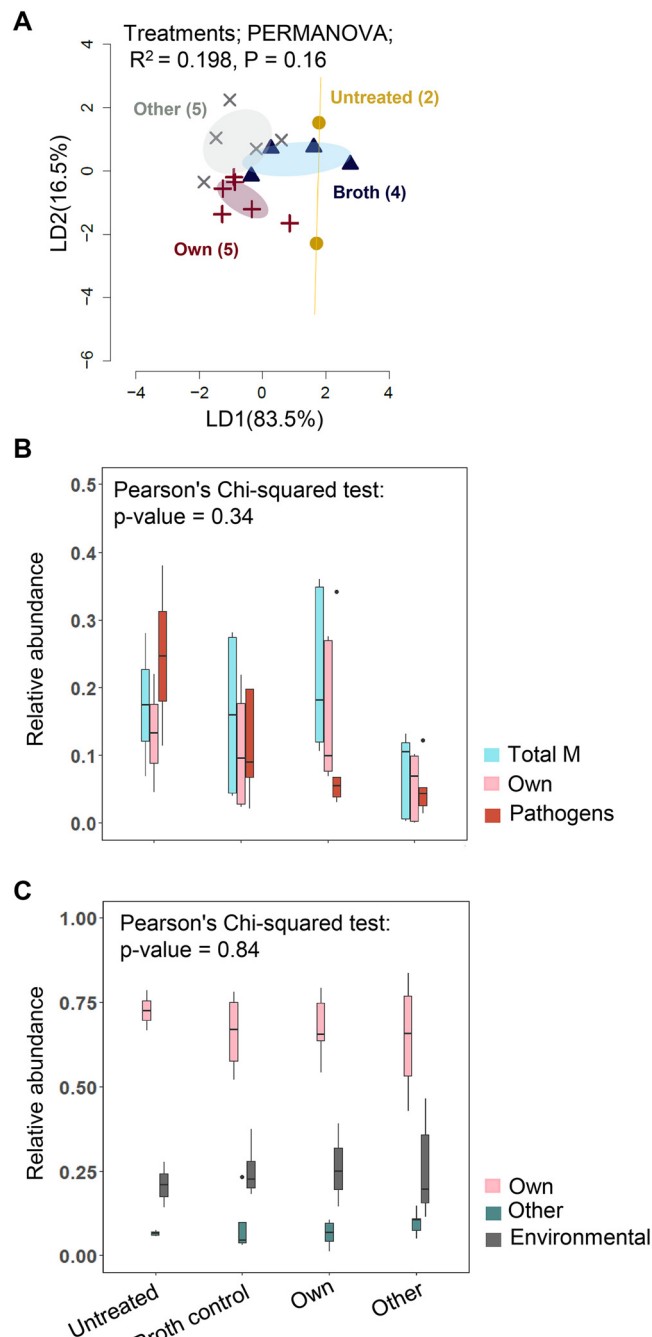

**FIG 5** Phyllosphere bacterial communities as a function of experimental treatments for the landrace Chakhao. (A) Linear discriminant (LD) plot showing the clustering of phyllosphere bacterial communities across treatments. The number in parentheses indicates the number of biological replicates in each treatment. Axis labels indicate the proportion of variation explained, and ellipsoids represent 95% confidence intervals. (B and C) Boxplots show the relative abundance of (B) all *Methylobacterium* (total M) versus own *Methylobacterium* versus pathogens and (C) own or other versus environmental *Methylobacterium* strains (total M; see Materials and Methods). Chi-squared tests showed the effect of treatment on the proportion of each group.

"other" *Methylobacterium* strains. Finally, we did not know the provenance of the *Methylobacterium* strains because it was not possible to prevent colonization by naturally occurring strains in the soil or air in field experiments. Hence, we could not distinguish between colonization by such environmental "own" *Methylobacterium* versus potential contamination across treatments.

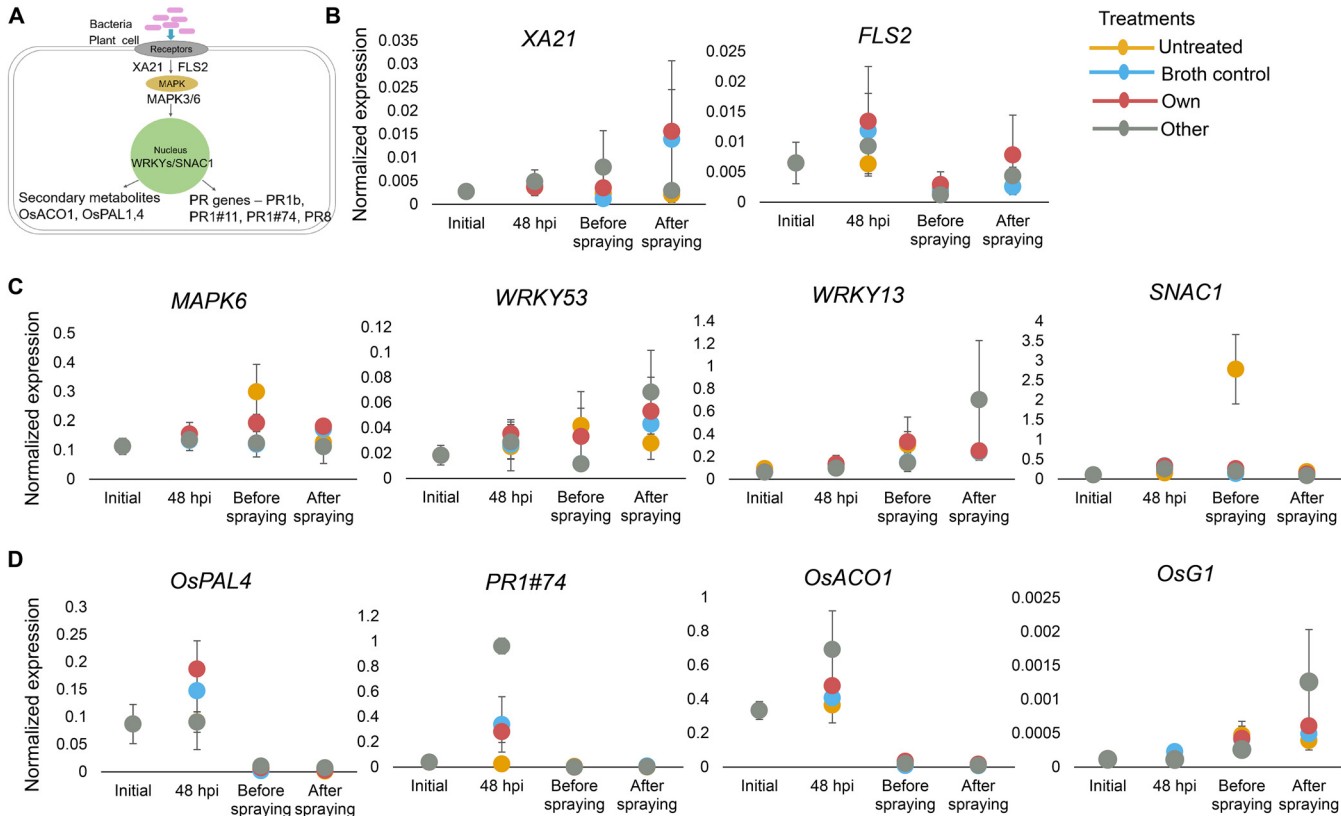

**FIG 6** Impact of Methylobacterium on early gene expression in host plants. (A) Schematic showing key signaling pathways and genes that regulate plant responses to abiotic and biotic stresses. (B to D) The expression of each gene before and after two rounds of bacterial inoculation (initial versus 48 h postinfection of seedlings; before versus after foliar spraying on 45-day-old plants) was normalized to the expression of internal control (*OsACTIN1*). Error bars represent the standard error from three biological replicates/treatments (three seedlings or plants).

Nonetheless, our results did not support the hypothesis that *Methylobacterium* inoculation influenced host fitness by altering phyllosphere colonization by pathogens, or more broadly, by altering the phyllosphere microbiome. Given that the "own" *Methylobacterium* strain seemed to be abundant on Chakhao plants in all treatments, we hypothesized that events soon after experimental inoculation by *Methylobacterium* may explain the observed benefits for the host. Hence, we tested whether *Methylobacterium* influences the early expression of important host plant signaling pathways.

**Early host plant gene expression following *Methylobacterium* inoculation.** Next, we determined whether *Methylobacterium*-rice interactions may have altered the expression of host genes involved in growth and defense, contributing to the observed phenotypes. We conducted a laboratory and greenhouse experiment where we exposed Chakhao seeds to the same four treatments as the field experiment and analyzed the expression of selected host genes. We destructively sampled seedlings at different stages, planting them in the greenhouse to monitor the early growth phase. Supporting our field results, we did not find a growth advantage for *Methylobacterium* during the early vegetative growth phase (Fig. S8).

We analyzed early gene expression from stored samples collected before versus after soaking seeds (before transplantation), and before versus after the foliar spray (soon after transplantation). We used qRT-PCR to measure the expression of genes known to play important roles in plant responses to pathogens, herbivores, and abiotic stresses (Fig. 6A), relative to a housekeeping gene. Most genes did not show a significant impact of treatment or a time × treatment interaction when considering pairwise differences before versus after inoculation (either for the first or the second bacterial inoculation; Table S6). The only exceptions were *SNAC1* and *PR1#74*, which showed significant or nearly significant impacts of time, treatment, and their interaction (*SNAC1*, ANOVA: time, $P = 0.07$, Cohen's f = 0.471; treatment,

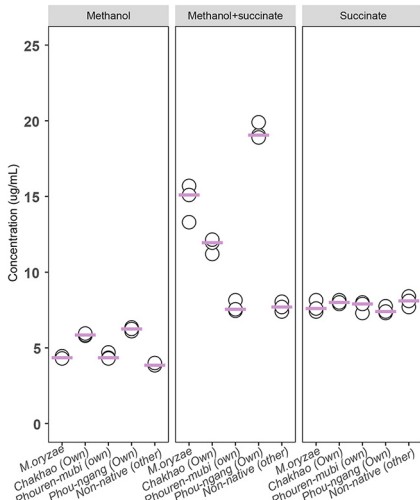

**FIG 7** Auxin (IAA) production in *Methylobacterium* strains. Plots show the concentration of IAA produced by each *Methylobacterium* strain (isolated from Chakhao, Phouren-mubi, Phou-ngang, and nonnative *Methylobacterium*; *M. oryzae* [*KACC11585*] used as a positive control; *n* = 3 biological replicates/strain) when grown on medium supplemented with various carbon sources (25 mM methanol, 25 mM methanol + 2 mM succinate, or 2 mM succinate). Each point represents a replicate culture; horizontal pink lines indicate the median.

$P = 0.05$, Cohen's f = 0.76; time x treatment, $P = 0.07$, Cohen's f = 0.72; *PR1#74*, ANOVA: time × treatment, $P = 0.07$, Cohen's f = 0.73). *SNAC1* is involved in drought resistance (20), and it was upregulated before spraying only in untreated plants (Fig. 6). This upregulation indicates that the plants experienced conditions similar to drought stress. Interestingly, the remaining treatments did not show a similar increase in *SNAC1* expression, hinting that the broth and *Methylobacterium* treatments might provide drought tolerance. *PR1#74* is usually upregulated under pathogen attack. In our case, it was only upregulated in the "other" *Methylobacterium*-treated plants. Although this supported the idea that plants respond to this *Methylobacterium* strain as if it were a pathogen, these results did not explain the positive impacts of the own *Methylobacterium* and broth alone. Other genes involved in pathogen response, *OSXA21* and *FLS2*, were not upregulated, which was expected because they specifically respond to the bacterial factor *Xoo* (*Xanthomonas oryzae*) and flagella, neither of which are present in the *Methylobacterium* groups to which our strains belong. Altogether, our gene expression analyses suggested that some host plant responses to environmental stresses may be altered by *Methylobacterium*. However, none of the observed effects (which were also weak) correlated with patterns of increased host fitness due to the own *Methylobacterium* and, therefore, could not explain the broad fitness effects. Hence, the mechanistic basis of the fitness benefit of native *Methylobacterium* as well as the bacterial growth medium remains unclear.

**Auxin production by *Methylobacterium* strains.** Finally, we tested whether differential auxin production by the different *Methylobacterium* strains used in our study could explain their variable fitness impacts. We compared our results to a reference strain of *Methylobacterium oryzae* (KACC 11585) that encodes some of the relevant metabolic pathways and, hence, served as a positive control. Surprisingly, during growth in a culture medium with methanol or succinate alone, all three native *Methylobacterium* strains produced a comparable amount of auxin to the reference *Methylobacterium oryzae*. In the presence of both methanol and succinate, auxin production varied substantially across strains (Fig. 7). However, the amount of auxin produced was not correlated with the relative fitness benefit conferred on the host plant. For instance, the largest amount of auxin was produced in the presence of methanol, by the strain associated with Phou-ngang (Fig. 7). However, this strain had no impact on host plant growth or yield. Hence, overall, *in vitro* assays did not support the hypothesis that landrace-specific impacts on host fitness

could be explained by differences in the strains' ability to produce auxin that may influence plant growth.

## DISCUSSION

Our work represents one of the first field analyses of the impact of phyllosphere bacteria on their host plants. Previously, we showed that leaves of different rice landraces are colonized by phenotypically distinct strains of *Methylobacterium*, with preliminary greenhouse experiments indicating landrace-specific benefits for plant growth (19). These results suggested that host selection might have driven specific *Methylobacterium*-rice landrace associations in paddy fields and that this could be utilized for increasing rice yields. Our results from the field experiment described here support the broad hypothesis that phyllosphere *Methylobacterium* benefit their host and that the effect varies across landraces. In contrast to our earlier greenhouse experiments, the field study did not find a large impact of *Methylobacterium* on vegetative growth traits. However, we did not measure below-ground plant growth such as root biomass, and this would be an important avenue for future work. The divergent outcomes between the current work and prior greenhouse results could reflect the different timescales of the experiments (greenhouse experiments were terminated at 40 days), differences in the inoculation protocol (in the greenhouse experiment, seeds were sterilized before inoculation), or different environmental conditions, including the pool of available environmental microbes and multiple abiotic factors that cannot be controlled in the field. However, in the current study, we did find significant impacts of *Methylobacterium* on yield-related traits (~2 to 6-fold increase), including the number and fraction of filled grains. Remarkably, Chakhao plants inoculated with their native *Methylobacterium* strain produced ~1200 grains, which was substantially more than untreated plants (~600) or plants treated with the broth control (~900) or nonnative *Methylobacterium* (~700). Thus, for highly valued landraces such as Chakhao and Phouren-mubi, inoculation with *Methylobacterium* could be developed as a method to improve yield.

Importantly, these benefits seem to depend on the specific association between each landrace and the predominant *Methylobacterium* strain that it hosts. A potential cause of such host-specific benefits is that host genotype determines interactions with bacterial partners, as observed in many other plants (21). In our case, such differences might mean that varieties such as Phou-ngang were unable to establish a mutualism with *Methylobacterium*, e.g., due to differences in leaf architecture or surface chemistry. For instance, colonization of different tomato cultivars by *Salmonella enterica* varies depending on the presence of type-1 trichomes on tomato leaves (22). A second possibility is that the properties of the different strains might determine host fitness impacts. For instance, the Chakhao-associated strain might be generally more beneficial, compared to the strains selected for Phouren-mubi or Phou-ngang. However, our assay of auxin production did not support the idea that the most beneficial "own" strains were distinguished by excess production of plant growth-promoting factors. We note that this analysis was limited to a single metabolite measured *in vitro* rather than during plant colonization, and future work should measure bacterial production of other important hormones such as cytokinins. The strains might also differ in their ability to colonize different landraces under competition, as observed for phyllosphere *Methylobacterium* of *Arabidopsis* (23). In our case, strains identical or similar to the "own" *Methylobacterium* dominated the phyllosphere microbiome across all treatments. Prior work showed that only some *Methylobacterium* strains successfully colonized rice and increased grain filling (16). Mirroring our results, this study also found that some strains either had no impact on or reduced rice growth or grain filling. Importantly, our nonnative "other" *Methylobacterium* strain appears to be harmful to all landraces, though the mechanisms underlying this effect remain unclear. We note that, due to logistical limitations, we only included one "other" and "own" strain in our study. In future work, it will be important to include more strains of each category to be able to generalize the effect of native versus nonnative bacteria. We also note the possibility that the "own" strains that we identified as native strains colonizing Phouren-mubi and Phou-ngang (19) were perhaps only transiently dominant and not consistently associated with these landraces, or that their

association is relatively recent. Lastly, variability in the impact of bacteria on host fitness may be driven by more complex interactions between multiple causal factors. For example, genetic differences across landraces may have led to divergent host-imposed selection on their bacterial associates, such that *Methylobacterium* associated with Phouren-mubi and Phou-ngang evolved to confer fitness benefits that would only be visible under specific environmental conditions. Indeed, leaf nutrients and soil conditions together influence the phyllosphere microbial community across different genotypes of rice plants (24). Similarly, the degree of prior adaptation to the local environment may also influence host-rice interactions. Phou-ngang is typically not cultivated in the region where our field site was located; perhaps it does benefit from its native *Methylobacterium* strains when cultivated in other areas. Finally, if gene content or phenotype is conserved across closely related strains or species, the phylogenetic position of the bacterial strains may also impact their interactions with plants. To distinguish between all these hypotheses, future work must analyze the temporal and spatial robustness of rice phyllosphere communities and conduct more extensive field trials to screen factorial combinations of rice landraces and *Methylobacterium* strains under diverse growing conditions. We hope that such analyses will improve our ability to predict which strains may or may not enhance host plant fitness.

Strikingly, in both Phouren-mubi and Chakhao, we found that *Methylobacterium* significantly reduced grain sterility (~20%) compared to control plants (~50%). What mechanisms might underlie these strong impacts? A recent meta-analysis shows that the effect size of beneficial microbes on plant health tends to be greater under stress (25). In Manipur, 2018 was a drought year, with the worst water shortage occurring during the reproductive phase, causing a severe loss in yield across the region (personal communication with farmers). Our field experiment might have been partially affected by the drought, although we tried to mitigate the effects (see Materials and Methods). Prior work shows that grain filling in rice is sensitive to drought because drought induces excess ethylene production (a stress hormone) that regulates the grain filling rate (26–28). Interestingly, bacteria such as *Pseudomonas putida* and *Pyrococcus horikoshii* can reduce stress-induced ethylene levels by secreting 1-aminocyclopropane-1-carboxylic acid deaminase, which breaks down the precursor of ethylene (reviewed in reference (29)). Similarly, *Methylobacterium fujisawaense* reduces ethylene levels in Canola, enhancing plant growth (30), and *Methylobacterium* sp.2A isolated from the rhizosphere of potato plants improved plant growth under adverse conditions (31). Thus, we speculate that in our experiment, the impact of *Methylobacterium* was at least partially mediated by the regulation of ethylene levels in response to drought stress. Future analyses are required to explicitly test this hypothesis.

A curious and unexpected result of our experiments is the strong beneficial impact of soaking seedlings and foliar spraying with the bacterial growth medium ("broth"). This effect could potentially be explained by methanol, which we added to the broth as a carbon source for bacterial growth. During plant growth, methanol is released as a by-product of cell wall remodeling due to the activity of pectin methyl esterases (PMEs). The application of methanol on leaves leads to upregulation of PME expression, resulting in plant growth promotion, as observed in tobacco and *Arabidopsis* (32). Another field experiment also indicated that foliar spray of 30% methanol enhances colonization and cytokinin production by *Methylobacterium*, increasing the growth of cotton and sugarcane (33), although this study had a small sample size. However, our growth medium contained very little methanol (~0.125%; Table S1). Further, broth and all treatments that included the growth medium tended to reduce growth-related traits in our experiments. Hence, it is unlikely that, in rice, the impact of the broth is similar to the growth-promoting effects observed in previous studies. We did observe a very strong effect of broth on yield-related traits in two landraces. What could explain this effect? One possibility is that the inoculation of broth enhances colonization by beneficial environmental microbes was not strongly supported by our data because we did not observe a significant difference in the phyllosphere microbiomes of untreated and broth-treated plants. However, it is possible that the microbiome changed at a critical early stage or in a different plant site such as the rhizosphere,

which we did not sample. Alternatively, our low sample sizes may have limited our ability to determine differences in phyllosphere microbiomes. Finally, the broth includes trace metals that can alter host plant gene expression at low concentrations (34, 35), potentially increasing grain yield. This is an intriguing possibility that should be explored further to understand the physiological mechanisms and potential agricultural applications. Finally, as mentioned in the results section, we likely underestimated the impact of bacteria on plant growth because of more nutrient availability in the broth control. Hence, we suggest that future work should use a spent filtered growth medium as a control.

In summary, our work highlights an important step in understanding and developing host-bacterial interactions for agriculture applications. Our results support a growing body of work showing that phyllosphere microbiomes are strongly shaped by local filtering imposed by host plants and that the fitness effects of bacteria on their hosts are not generalizable. We further show that several potential mechanisms, including altered phyllosphere microbiome composition, reduced pathogen abundance, *in vitro* auxin production, and early host plant gene expression, do not explain the variable impacts of the different bacterial strains and highlights the need for further work. Nonetheless, we show that inoculating traditionally cultivated rice varieties with specific *Methylobacterium* strains is a promising targeted method to mitigate the impact of environmental stress on grain filling and enhance productivity.

## MATERIALS AND METHODS

**Bacterial strains and culture conditions.** In 2016, we isolated hundreds of *Methylobacterium* strains from the phyllosphere of multiple rice landraces in northeast India (19). In this study, we analyzed the impact of the association of four of these strains on host rice plants in the field, choosing strains that were dominant in the respective host landrace. Three isolates were associated with rice varieties grown in the state of Manipur: strain CKPL1 (GenBank accession no. MN982816) from rice landrace Chakhao, strain PML2 (GenBank accession no. MN982833) from Phouren-mubi, and PNL1 (GenBank accession no. MN982835) from Phou-ngang. The fourth, DKS6 (GenBank accession no. MN982770), was isolated from Deku, a rice landrace in Arunachal Pradesh. Based on 16S rRNA sequencing, CKPL1 appears to be closely related to *Methylobacterium komagate*; PML2 to *Methylobacterium radiotolerans*, PNL1, and DKS6 to *Methylobacterium salsuginis* (the latter two strains are now reclassified as the genus *Methylorubrum*) (36). To prepare bacterial inoculum, we inoculated 40 $\mu$L of *Methylobacterium* glycerol stocks in 300 mL Hypho supplemented with 60 mM methanol and 2 mM succinate as the carbon source and allowed growth for 48 h at 30°C in a shaking incubator (see Table S1 for medium composition; (37)). The final optical density at 600 nm ($OD_{600}$) ranged from 1 to 1.3 depending on the growth of the *Methylobacterium* strain used.

**Field experiment.** We identified a plot of farmland in Chingarel village of the Imphal East district of Manipur, a part of a larger area cultivated by a local farmer. Usually, the farmer cultivates 3 to 5 different rice varieties simultaneously in the same field. However, in 2018 the farmer decided to only plant a commercially available high-yielding rice variety. We designed our experimental plot such that it was not close to these plants (Fig. S1A). The experiment was conducted from June to December 2018 (Fig. 1), coinciding with the usual rice-growing season in this region (Fig. 1A). A month before planting, the field was prepared, as usual, using a hoe to plow the soil and raising mud boundaries to separate the treatment blocks and without the addition of any fertilizer to the experimental plot (Fig. S1A).

We obtained seeds of the three experimental rice landraces (Chakhao, Phouren-mubi, and Phou-ngang) from the preceding year's harvest from the same farmer. For each landrace, we divided the seeds into four treatment groups: (i) untreated, treated only with water, (ii) broth control, treated with sterile bacterial growth medium, and (iii) own, treated with the respective native *Methylobacterium* strain isolated from the landrace, and (iv) other, treated with the nonnative *Methylobacterium* strain from Arunachal Pradesh. To ensure that we used viable seeds, we used excess seeds and allowed them to germinate for 4 days while wrapped in a wet cloth. We prepared the bacterial cultures and soaked only germinated seeds in their respective inoculum for 48 h (for instance, 300 mL culture for 300 germinated seedlings in a flask; Fig. S1B). We then transplanted the soaked seedlings into the field, placing 4 seedlings together to constitute one "plant," and using a wooden frame with 1 ft × 1 ft squares to ensure sufficient spacing between plants (Fig. S1C). For each landrace and treatment combination, we transplanted 30 plants. Subsequently, we analyzed only 15 of them (Fig. 1B) to minimize the impacts of cross-contamination across neighboring plants. These rice landraces are typically cultivated as follows: seedlings are transplanted in wet soil, and then the field is allowed to flood under natural rainfall (wetland cultivation) until the ripening stage. However, in 2018 Manipur faced a severe drought, with very little rainfall during the plants' reproductive phase (October onwards). To mitigate drought stress, we irrigated the experimental plot with purchased water in late October. All farming operations except the bacterial treatments were carried out by the farmer.

Forty-five days after transplantation of seedlings (~6 to 8 green leaves per plant), we sprayed each rice plant with 10 mL of fresh inoculum (water, broth control, or broth with own or other *Methylobacterium* strains) (Fig. 1A). To test whether the sprayed bacteria successfully colonized the plants, we collected flag leaves from three plants per treatment, before and after spraying. We isolated *Methylobacterium* by imprinting the leaves on Hypho minimal agar with 0.5% methanol and identified the strains by sequencing the 16S rRNA gene as described earlier (19). As the rice plants grew, we measured traits such as plant height, flag leaf width, and length, and the number of tillers and panicles (38) at three different time points (Fig. 1A). Finally, in early December, we collected all the grains produced by each plant and transported them to the laboratory in airtight plastic bags to measure the total number of grains and total grain weight per plant (38). During transportation, some plants were infested by mold and therefore we had a slightly lower sample size than expected for the yield measurements.

**Determining phyllosphere microbiome composition.** To determine the phyllosphere microbiome as a function of experimental treatments, we collected flag leaf samples at the seed ripening stage in early December (*n* = 2 to 6 flag leaves/plant/treatment/landrace). To obtain epiphytic bacteria from the leaf surface, we used a previous protocol with some modifications (39). Briefly, we placed the cut leaf (3 to 4 cm length) into 1 mL of 1× Redford buffer (1 M tris HCl; 0.5 M EDTA; 1.24% Triton) for 5 h with periodical vortexing. We removed the leaf tissue and used the supernatant (containing bacteria dislodged from the leaf surface) for DNA extraction using a Qiagen DNeasy blood and tissue kit, with slight modifications to the standard bacterial protocol as follows. We centrifuged the leaf wash at 10000 rpm for 25 min. To the pellet, we added 20 $\mu$L proteinase K for 10 to 15 min at 56°C and added 200 $\mu$L of lysis buffer at 56°C for 2 h. After this, we followed the manufacturer's protocol without modification. We quantified the DNA in each sample using the Qubit 3 fluorometer (Invitrogen, ThermoFisher Scientific Inc.). We then amplified the V3-V4 hypervariable regions of the 16S rRNA using standard Illumina primers: F5'TCGTCGGCAGCGTCAGATGTGTATAAGAGACAGCCTACGGGNGGCWGCAG3' and R5'GTCTCGTG GGCTCGGAGATGTGTATAAGAGACAGGACTACHVGGGTATCTAATCC3' (underlined bases indicate the adapter overhang). We blocked the amplification of plant mitochondrial and chloroplast DNA by adding PNAs (polypeptide nucleic acid) in our 16S PCR, as described previously (40). We sequenced amplicons on the Illumina Miseq platform (300 × 2 paired-end reads).

To analyze the microbiome, we used the DADA2 workflow to filter out reads with low quality (Q < 30) and generate ASV (amplicon sequence variant) tables (41). Briefly, reads with 100% sequence identity were retained and assigned taxonomy using the Silva reference database (training set v138.1) (42, 43). After filtering, we obtained an average of 40000 reads (range 32000 to 80000) per sample. To determine successful colonization by own and other *Methylobacterium* strains, we binned all *Methylobacterium* into three groups: identical or very similar (at least 97% sequence identity) to the experimentally inoculated "own" or "other" strains; or "environmental" *Methylobacterium* (*Methylobacterium* ASVs that were not classified as own or other and were, therefore, likely acquired from the environment). We calculated the relative abundance of each group, either as a fraction of the entire microbial community (i.e., all microbial reads) or as a fraction of all *Methylobacterium* reads. Similarly, we determined the relative abundance of the most common rice pathogens (*Burkholderia*, *Pseudomonas*, and *Xanthomonas*) (44, 45) using sequence identity (97 to 100%) to the 16S rRNA sequences from the NCBI database (GenBank accession no. MN400211.1, FJ151352.1, HM747119.1, respectively).

**Greenhouse experiment to test the impact of *Methylobacterium* on gene expression in host seedlings.** We tested whether inoculation of *Methylobacterium* leads to a change in PAMP (pathogen-associated molecular pattern) triggered immunity genes or stress-related genes in the early stages of the host plant. We conducted a greenhouse experiment using the same inoculation protocols as described for the field experiment but focusing on the rice landrace Chakhao. Based on earlier studies, we chose representative host genes that are important for host responses to various stresses, including the membrane receptor kinases *OsXA21* (specific to *XOO* factor of *Xanthomonas*; (45)) and *OsFLS2* (specific to flagella, (46)); the mitogen-activated protein kinase *OsMAPK3/6* (involved in plant defense, (47)); transcription factors *OsWRKY13*, *53*, *62*, and *76* (involved in plant defense; (47)) and *SNAC1* (drought stress; (20)); pathogenesis-related proteins *PR1b*, *PR1-11*, *PR1-74*, *PR8* (48); and genes involved in the production of secondary metabolites like *OsPAL1,4* (lignin biosynthesis, (49)) *OsACO1* (ethylene regulation, (50)) and *OsG1* (involved in plant defense and development, (51)).

After soaking seedlings in the respective bacterial cultures (or controls), we transplanted them into small pots (2 germinated seedlings per pot) maintained under flooding. We grew all plants in the greenhouse under natural light with 70% relative humidity. At four different time points (immediately before and after soaking seedlings in bacterial inoculum; and before and 24 h after spraying young plants with inoculum), we collected three replicate tissue samples (three seedlings or leaves from three plants) for each treatment (Fig. S2) and stored them at −80°C. We extracted total RNA from a single seedling or leaf using TRIzol (Invitrogen) as per the manufacturer's instruction. We used 2 $\mu$g RNA for cDNA synthesis using a RevertAid cDNA synthesis kit (ThermoFisher) and measured gene expression using qRT-PCR (Bio-Rad CFX system). For each biological replicate, we tested three technical replicates and normalized CT values for each target gene against *OsACTIN1*. Primers used for amplification are listed in Table S2. We then calculated the normalized expression of each gene as $2^{-\Delta Ct}$, where $\Delta C_t = C_{t(gene)} - C_{t(actin)}$.

**Testing auxin production by focal *Methylobacterium* strains.** We tested whether our four *Methylobacterium* strains differed in their ability to produce auxin (IAA, indole acetic acid), using the Salkowski colorimetric assay (52). We grew each strain (5 $\mu$L inoculum from glycerol stocks) in 20 mL Hypho broth supplemented with either 25 mM methanol, 25 mM methanol with 2 mM succinate, or 2 mM succinate alone (3 independently inoculated tubes per strain). We added 5 mM L-tryptophan to the growth medium (substrate for IAA production) and incubated cultures at 28°C with shaking at 150 rpm for 5 days. We measured OD$_{600}$ of each culture and

diluted it with fresh sterile Hypho as necessary so that all cultures had a similar cell density (OD = 1). After centrifugation at 10,000 rpm for 10 min, we took 1 mL of the supernatant, added 1 mL of Salkowski reagent, and incubated for 1 h in the dark at room temperature. We measured IAA production by measuring color production at 530 nm (spectrophotometer Hitachi UH5300) and estimated the amount of IAA produced using a standard curve made from serial dilutions of synthetic IAA (Himedia) in Hypho broth.

**Statistical analysis.** We analyzed and visualized data using R (53). For all plant traits, we detected potential influential points using the function influential.measures in base R, and we report statistical analysis both with and without these influential points. We used two-way ANOVAs to test the effect of treatment and host landrace on each trait, using a generalized linear model (GLM) with normal error distribution, and Tukey's honestly significant difference (Tukey's HSD) to account for multiple comparisons in the R package 'multcomp' (54).

To test the impact of bacterial treatment on the phyllosphere microbiome, we used PERMANOVA (permutational analysis of variance) using the function 'Adonis' in the package 'Vegan' (55). To visualize sample clustering based on bacterial community composition, we calculated Bray-Curtis distances between samples and performed a canonical analysis of principal coordinates based on discriminant analysis (CAPdiscrim) (56) using the R package 'Biodiversity R' (57). We tested for significant clustering and estimated classification success by permuting the distance matrix 1000 times and estimating the probability of finding the observed differences by chance. We plotted the two dominant linear discriminants (LD) to visualize clusters. For each cluster, we drew ellipses reflecting 95% confidence intervals using the function 'Ordiellipse' in the R package 'vegan' (55).

For the qRT-PCR data, for normalized expression levels of each gene, we used generalized linear models (GLM) with Tukey's honestly significant difference (Tukey's HSD) to account for multiple comparisons. We carried out ANOVAs to test the effect of time and treatment on gene expression. For visualization of the data sets, we used the package 'ggplot2' (58).

**Data availability.** Raw sequence data for microbiome are available in the Sequence Read Archive (SRA) of the National Center for Biotechnology Information (NCBI) under Bioproject PRJNA824289. All the data supporting the findings are given in the paper in Supplemental File 1 and 2.

## SUPPLEMENTAL MATERIAL

Supplemental material is available online only.
**SUPPLEMENTAL FILE 1**, XLSX file, 1.7 MB.
**SUPPLEMENTAL FILE 2**, PDF file, 1.2 MB.

## ACKNOWLEDGMENTS

We thank farm owner W. Ibobi Singh for access to the field plot and farming operations; S. Chandrakanta Singh for assistance with fieldwork; L. Shanjukumar Singh (Manipur University) for access to laboratory facilities during fieldwork; and the Korean Agriculture Culture Collection for providing reference *Methylobacterium oryzae* strain KACC 11585. We also thank Shreya Vichare, Rahul Keshav, Pranjal Gupta, and Manjunatha K Reddy for their help with data collection; Harshith C.Y., Sujith T.N., and Vivek Hari Sundar for help with qRT-PCR; and Rittik Deb and Nitish Malhotra for help with microbiome and sequence analysis.

We acknowledge funding and support from the Department of Biotechnology (DBT grant no. 358 NER/AGRI/24/2013), the National Centre for Biological Sciences (NCBS-TIFR), and the Department of Atomic Energy, Government of India (project Identification no. RTI 4006).

P.S. conceived, designed, and conducted experiments, analyzed data, and drafted the manuscript. P.V.S. designed experiments and provided reagents and laboratory facilities. D.A. conceived and designed experiments, directed analysis, wrote the manuscript and acquired funding.

We declare no conflict of interest.

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
