## [Reviewer comments · Microbiology Spectrum]

Microbiology Spectrum

Impact of phyllosphere *Methylobacterium* on host rice landraces

Pratibha Sanjenbam, P Shivaprasad, and Deepa Agashe

Corresponding Author(s): Deepa Agashe, National Centre for Biological Sciences

Review Timeline:

Submission Date:	March 7, 2022
Editorial Decision:	April 1, 2022
Revision Received:	May 18, 2022
Accepted:	June 24, 2022

Editor: Erik Hom

Reviewer(s): Disclosure of reviewer identity is with reference to reviewer comments included in decision letter(s). The following individuals involved in review of your submission have agreed to reveal their identity: Victoria Calatrava (Reviewer #1); Steffen Kolb (Reviewer #2)

Transaction Report:

DOI: <https://doi.org/10.1128/spectrum.00810-22>

April 1, 2022

Prof. Deepa Agashe
National Centre for Biological Sciences
Bangalore
India

Re: Spectrum00810-22 (Impact of phyllosphere Methylobacterium on host rice landraces)

Dear Prof. Deepa Agashe:

Thank you for submitting your manuscript to Microbiology Spectrum. Three reviewers have considered your work and their comments are appended. All suggest varying degrees of revisions, that I invite you to carefully consider making in a revised manuscript.

One of the more pointed criticisms raised by a reviewer was that there was little scientific novelty to improve our understanding of Methylobacterium interactions with rice plants or bigger-picture outcomes. Also raised was that the manuscript was long with a conclusions section that did not highlight the novelties or importance of the study's outcome. While "novelty" is not a criterion for Microbiology Spectrum, I would encourage you to try and better situate your work in a broader context and to tighten up the message of your manuscript. In your resubmission cover letter, please make sure to summarize the changes you made and how you addressed the reviewer's major criticisms.

Link Not Available

Sincerely,

Erik Hom

Journals Department
Reviewer comments:

Reviewer #1 (Comments for the Author):

Plants have long co-existed and co-evolved with benefiting bacteria. Despite its great significance in crop production, many details about the underlying specificity of host-microbe interactions are still unclear. This work therefore addresses an important question concerning the impact of "native" versus "non-native" bacterial species from the genus *Methylobacterium* (a major genus in the phyllosphere) on different rice landraces. The authors tested how different methylobacterial strains impacted plant growth and productivity in the field, phyllosphere microbiome composition and plant gene expression. They also examined the ability of these bacteria to produce auxin. The authors found variable results dependent on specific landrace rice species used and overall, these results were well presented and discussed. Limitations of their study are also pointed out, along with suggested follow-up studies to address missing gaps generated over the course of this work. I do not have any major issues, but do have a few minor suggestions and comments.

- Lines 501-502: Add references for genes involved in plant response to pathogens, herbivores, abiotic stresses.

- In Figure 7, remove one of the scales for y axis. I suggest removing OD and keeping concentration. Correct the units ($\mu\text{g/ml}$?).

- Line 272: Were the cells diluted after growth and prior to auxin determination? If so, the details for how these measurements were determined was not clear to me.

- The authors speculate that at least part of the effect of *Methylobacterium* inoculation is due to the regulation of ethylene levels in plants in response to stress by means of ACC deaminase activity. Do the specific isolates used in this study have or lack ACC deaminase? If no genomic data is available, are they present in the most closely-related species? Are there differences in the presence or absence that could explain the differences observed?

- Although the authors address that one of the limitations of this study is the use of only one "non-native" isolate, they do not address potential effects or differences of using phylogenetically different *Methylobacterium* isolates. For instance, according to authors, the isolate native to Phou-ngan (which was not as beneficial as other native isolates) and the one used as "non-native" are more closely related to each other than to the rest of the native ones used. This raises the possibility that this non-native isolate is also not as beneficial (at least in this environment). I would have found it more interesting to use one of the 'native' isolates as a 'non-native' partner to another rice landrace to determine whether the positive effect of such an isolate disappears when applied to a different landrace. Nevertheless, I understand this substantially increases the complexity of the experiments and study design; the scope of what was done here is acceptable to me, but it would be nice if the authors could discuss these cross-compatibilities from a phylogenetic perspective a bit more in their discussion.

Reviewer #2 (Public repository details (Required)):

The DNA sequence data. A repository is stated but NOT the accession numbers.

Reviewer #2 (Comments for the Author):

The study presented is interesting since it addresses microbe plant interaction in rice landraces and not in conventional high yield varieties. However, the study lacks important plant growth parameters, such as belowground biomass of roots. This is needed to understand what exactly the effects of added *Methylobacterium* strains were. Likely, resource allocation within plants occur. The mechanisms why some strains lead to aboveground biomass and further phenotypic changes remains unresolved.

The gene expression data on the chosen plant enzymes are helpful. However, as the reviewer read the panels of Fig 6, four condition/plant stages were investigated - hence, linking them with lines is NOT appropriate. It is also NOT evident if statistical significant differences were found?! Also the number of biological treatments is for a typical plant experiment low (i.e. often 5-10 biological replicates)

There should be accession numbers for all sequences - Not given in the manuscript draft?

The discussion section is unnecessary long as the whole manuscript, and it lacks a clear concise conclusion what is novel and not only confirmatory to what is known on the phyllosphere microbiome or the effect or nature of interaction of *Methylobacterium* with rice and its phyllosphere microbiome.

The authors should stay consistent over the manuscript - just use 'bacterial microbiome' OR 'bacterial community', would improve readability.

Figure 7. Here the authors investigated physiological potentials of some re-isolated strains. If the reviewer got that correct, this should also be considered in the text. It is NO proof for an in situ activity then.

The last paragraph of the introduction contains some kind of summary. this is not needed and redundant and should be deleted.

Reviewer #3 (Public repository details (Required)):

16S sequencing data

Reviewer #3 (Comments for the Author):

The authors tested the effect of native vs non-native *Methylobacterium* inoculation on three different rice varieties in a field trial. They measured a large number of different traits and found that the results varied between variety and strain inoculation, indicating that results are not generalizable across *Methylobacterium* strains and rice landraces. They then go on to explore different mechanisms that might explain these variations but encounter mostly negative results. Due to the setup of the experimental field design, the interpretation of the results is limited. Apart from the "other" strain, *Methylobacterium* strains have been tested on a single landrace. However, the authors are upfront about the limitations and mostly careful with their interpretation. The manuscript is largely well and clear written. There are a few clarifications (especially regarding figure design) that would improve the manuscript and help in the interpretation of the results.

Comments:

1. Line 13: "dominate" might be a strong word, I would suggest rather "abundantly found" or similar
2. Line 22: "than the bacterial growth medium on its own"
3. Line 25: remove "clear"
4. Line 42: "inoculation with some bacteria..." as you do not show that these bacteria directly increase rice grain production and especially the non-native strain itself reduces plant health.
5. Line 142-143: What OD can the different strains reach within these 48 hrs? Are they stationary at that point and are similar in yield?
6. Which landrace was grown on the plot beforehand? Was it one of the tested landraces?
7. Line 180: 45 days after transplantation -> what is the growth stage at that time point?
8. Line 185: I assume you added an antifungal compound to the plates?
9. L189-199: Have you tested whether all bacteria are reliably dislodged from the leaf cuttings?
10. L215: How deep did you sequence?
11. L217-231: From your materials and methods section, I gather that you did the analysis at 97% OTU level rather than at the level of ASV. Could you please specify?
What is the reasoning behind clustering *Methylobacterium* spp at 97% sequence identity to check whether your inoculated strains can be detected on rice plants but using 100% sequence identity when looking at pathogen presence? I would be better to use ASVs to make an assumption about the presence of your inoculated strains as 97% sequence identity is generally only species level.
- I am also a bit puzzled by your use of Chi-square tests in e.g. in Fig 5. Since you are working with compositional data from sequencing, other tests would be more appropriate. I would encourage you to use e.g. edgeR or DESeq2 for checking for differences in an OTU between treatments.
12. You write that you use GLMs for data analysis. What link function did you use? How did you control for multiple testing of the large number of traits you looked at?
13. Fig. 3A: I suggest using continuous y-axes for all landraces to reduce confusion, especially since there does not seem to be a need for split y-axes. Also, since this shows difference in growth between T3 and T1 and growth rate does not seem to be constant (based on Fig S3), I would suggest to show absolute difference in size rather than growth rate.
14. Fig. 3B: Why are there less samples than in A? Did you not determine grain yield for all plants? What is the difference between Fig S5A and Fig 3B? For me it looks like the same data is depicted but significance asterisks differ between the two.
15. L 356: Based on Fig S4 only total plant height tends to be decreased relative to untreated. Relative to the broth control most measurements are either similar or higher for the "own" treatment and only the "other" treatment might be lower. Thus, the conclusion that *Methylobacterium* reduce vegetative growth across land races seems not in line with the data.
16. For your graphs, you generally indicate significant differences relative to the broth control only. As the broth control is not the left most treatment and not labelled as "the" control this is a bit confusing because one would thus assume that e.g. own vs untreated is not significantly different when there is no significant asterisk indicated. Can you please comment?
17. Fig. 4: I would recommend to show the differences as (log)-fold-changes rather than in absolute values. It is difficult to compare for example a difference of 5 in flag leaf length to a difference of 300 in number of filled grains since the scales between these differ strongly. These graphs do not give easy access to the effect sizes. Also for "own" you compare two different bacterial treatments, whereas for "other" it is the same. It would be helpful to have this information in the legends. Furthermore, Table S4 and not Table S3B should be referenced in the text along Fig. 4.
18. L392: "can be beneficial"
19. The recovered *Methylobacterium* strains that you had from plating, did you only pick different morphologies or how did you quantify these? From Fig S6 it looks like you have for example many pink colonies on plant 3 of untreated Phou-ngang or in the own "Chakhao" but Table S5 mentions only few total isolates for the different ones. Also, are the "own" and "other" here based on 100% sequence identity of the 16S?
20. L439: all your effects could also be explained by effects on other microbes present in the community and not by direct interactions with the host. Furthermore, they could for example also be caused by differences e.g. in pH or other parameters of your growth medium that is applied together with your strains. Have you checked for differences between the strains? Or what

happens when you wash bacteria before you inoculate them?

21. Fig. 5: Have you tested whether rel. pathogen abundance differs between own and other? For me, it looks like it is similar between the two, thus I would not specifically mention that pathogen proportion was lowest in "other" treatment. For panel C please label y axis to make clear that we are only looking at the relative abundance of the total M.

22. For the microbiome analysis when you specifically look at the Methylobacterium proportion, you group them into "own", "other" and "environmental". How do you treat then an "own" of e.g. Phouren-mubi on Phou-ngang? Would this be counted as "own" or not? I am wondering whether on Phou-ngang the inoculated strain itself is just less fit and the environmental could be a different strain.

23. L 465: I would claim that the relative abundance of "own" in Phouren-mubi does not look really lower than in Chakhao. Did you do an analysis?

24. L521: Did you check specifically whether flagella are absent in your Methylobacteria? Because there are Methylobacteria that contain flagella.

25. In the introduction you go quite a bit into detail about cytokinin production by Methylobacterium spp but this aspect is then neither tested nor further discussed as a possible mechanism. Why did you test auxin and not cytokinin production?

26. Table S1: There seems to be a mistake in the amount of succinate supplied per plant

Staff Comments:

Preparing Revision Guidelines

Please return the manuscript within 60 days; if you cannot complete the modification within this time period, please contact me. If you do not wish to modify the manuscript and prefer to submit it to another journal, please notify me of your decision immediately so that the manuscript may be formally withdrawn from consideration by Microbiology Spectrum.

We thank the reviewers for their constructive suggestions. Our responses to reviewer comments are highlighted in bold. Line numbers in the responses refer to the clean, revised manuscript (Manuscript_text_pdf_linenumbers.pdf).

Editor's comments

One of the more pointed criticisms raised by a reviewer was that there was little scientific novelty to improve our understanding of *Methylobacterium* interactions with rice plants or bigger-picture outcomes. Also raised was that the manuscript was long with a conclusions section that did not highlight the novelties or importance of the study's outcome. While "novelty" is not a criterion for *Microbiology Spectrum*, I would encourage you to try and better situate your work in a broader context and to tighten up the message of your manuscript.

In the abstract as well as the "importance" section, we clearly state that this is one of the first studies to demonstrate the fitness consequences of *Methylobacterium* inoculation of plants under field conditions. In the first paragraph of the Introduction, we outline prior work on plant microbial interactions. In the second paragraph of the Introduction, we highlight important results from previous work on plant–*Methylobacterium* interactions, and discuss (lines 91-98) that relatively little is known about what happens under field conditions, and whether there are specific local associations between host plants and material strains. We reiterate this in the first paragraph of the discussion section; we discuss our results in light of prior work on mechanisms underlying *Methylobacterium*–plant interactions in the body of the discussion section; and in the last paragraph, we again highlight the key conclusions from our work about host-specific interactions and the need to account for these while developing agricultural applications. As it turned out, none of the mechanisms that we tested had clear explanatory power, but not due to lack of effort or poor design. These negative results are themselves interesting, and point to clear avenues for future work that we have discussed. We have now added a sentence highlight this in the concluding paragraph (lines 685–689).

As some reviewers mentioned, one of the strengths of our paper is that it is well written, and that we have extensively discussed the limitations of our work and nuances of interpretation. Many of these points were added during previous rounds of review, and we think the manuscript was greatly improved as a result. Though we tried hard during this round of revision, we were not able to find large sections that could be removed without loss of flow and important points.

Reviewer #1

Plants have long co-existed and co-evolved with benefiting bacteria. Despite its great significance in crop production, many details about the underlying specificity of host-microbe interactions are still unclear. This work therefore addresses an important question concerning the impact of "native" versus "non-native" bacterial species from the genus *Methylobacterium* (a major genus in the phyllosphere) on different rice landraces. The authors tested how different methylobacterial strains impacted plant growth and productivity in the field, phyllosphere microbiome composition and plant gene expression. They also examined the ability of these bacteria to produce auxin. The authors found variable results dependent on specific landrace rice species used and overall, these results were well presented and discussed. Limitations of their study are also pointed out, along with suggested follow-up studies to address missing gaps generated over the course of this work. I do not have any major issues, but do have a few minor suggestions and comments.

Thank you for these positive comments.

•Lines 501-502: Add references for genes involved in plant response to pathogens, herbivores, abiotic stresses.

The references were given in lines 245-251 (methods section).

•In Figure 7, remove one of the scales for y axis. I suggest removing OD and keeping concentration. Correct the units ($\mu\text{g/ml}$?).

As suggested, we have retained only the IAA concentration axis. The units for IAA are correct.

•Line 272: Were the cells diluted after growth and prior to auxin determination? If so, the details for how these measurements were determined was not clear to me.

Yes, the cells were diluted, as described in lines 276-278. We have now also specified that the dilution was conducted so as to obtain a uniform OD of 1 for all cultures, before measurement.

•The authors speculate that at least part of the effect of *Methylobacterium* inoculation is due to the regulation of ethylene levels in plants in response to stress by means of ACC deaminase activity. Do the specific isolates used in this study have or lack ACC deaminase? If no genomic data is available, are they present in the most closely-related species? Are there differences in the presence or absence that could explain the differences observed?

This is an interesting point. However, we do not have genome sequences for our strains; as mentioned in the discussion, we suggest that this is grounds for further work. We did not want to speculate on this aspect too much, because there is a lot of variation in gene content across species and strains.

•Although the authors address that one of the limitations of this study is the use of only one "non-native" isolate, they do not address potential effects or differences of using phylogenetically different *Methylobacterium* isolates. For instance, according to authors, the isolate native to Phou-ngang (which was not as beneficial as other native isolates) and the one used as "non-native" are more closely related to each other than to the rest of the native ones used. This raises the possibility that this non-native isolate is also not as beneficial (at least in this environment). I would have found it more interesting to use one of the 'native' isolates as a 'non-native' partner to another rice landrace to determine whether the positive effect of such an isolate disappears when applied to a different landrace. Nevertheless, I understand this substantially increases the complexity of the experiments and study design; the scope of what was done here is acceptable to me, but it would be nice if the authors could discuss these cross-compatibilities from a phylogenetic perspective a bit more in their discussion.

This is also an excellent point. Although we don't feel very confident about the placement of our strains within the phylogenetic tree of *Methylobacterium*, we have mentioned the possibility of a phylogenetic signal also being important in determining differences between strains (lines 621-623).

Reviewer #2

(Public repository details (Required)): The DNA sequence data. A repository is stated but NOT the accession numbers.

The accession number is added in the manuscript and will be made public upon acceptance (line 712).

The study presented is interesting since it addresses microbe plant interaction in rice landraces and not in conventional high yield varieties. However, the study lacks important plant growth parameters, such as belowground biomass of roots. This is needed to understand what exactly the effects of added *Methylobacterium* strains were. Likely, resource allocation within plants occurs. The mechanism why some strains lead to aboveground biomass and further phenotypic changes remains unresolved.

We agree that it would be useful to measure additional traits such as root biomass to investigate potential trade-offs in plant growth parameters, and we had considered this. However, it was logistically challenging to handle and measure all the different traits for the large number of plants included in our study. In the case of Chakhao where we find a significant plant-bacterial interaction, it would be absolutely important and interesting to analyze other plant traits. However, this must be done in future work, and we have now mentioned this in the discussion (line 564-566).

The gene expression data on the chosen plant enzymes are helpful. However, as the reviewer read the panels of Fig 6, four condition/plant stages were investigated - hence, linking them with lines is NOT appropriate. It is also NOT evident if statistically significant differences were found?! Also, the number of biological treatments is for a typical plant experiment low (i.e. often 5-10 biological replicates)

We have now removed lines connecting points, as suggested. In most cases we did not find statistically significant trends, as was described in the text (lines 507-510) and in table S6. We respectfully disagree about the number of replicates (one of us is a plant molecular biologist)

– 3 biological replicates, each with 3 technical replicates, is the norm for qRT PCR studies.

There should be accession numbers for all sequences - Not given in the manuscript draft?

Accession numbers for the 16S sequence for all our focal bacterial strains is given in line numbers 132-135. The accession number for the microbiome data is added in line number 712.

The discussion section is unnecessary long as the whole manuscript, and it lacks a clear concise conclusion what is novel and not only confirmatory to what is known on the phyllosphere microbiome or the effect or nature of interaction of *Methylobacterium* with rice and its phyllosphere microbiome. **We understand this concern, but we are not sure exactly which points the reviewer would like to see removed. We also respectfully disagree that the discussion is unnecessary. It was expanded to address previous reviewers' concerns, and we believe that it is quite useful, especially for naïve readers. Hence, we have retained the points made in this section.**

The authors should stay consistent over the manuscript - just use 'bacterial microbiome' OR 'bacterial community', would improve readability.

We prefer to use both terms, since they are commonly used interchangeably in literature. It is also nice to not repeat the use of the same term frequently.

Figure 7. Here the authors investigated physiological potentials of some re-isolated strains. If the reviewer got that correct, this should also be considered in the text. It is NO proof for an in situ activity then.

In figure 7, we tested IAA production of the *Methylobacterium* strains used for field experiment; these were not re-isolated strains.

We have mentioned in line 548 that the IAA is indeed an *in vitro* assay and we mention in the discussion as a caveat that IAA production *in planta* is not known (line 593).

The last paragraph of the introduction contains some kind of summary. this is not needed and redundant and should be deleted.

We would prefer to retain this summary, as we think it helps guide readers to understand the key results and outcomes of our study. However, as suggested, we have now added a sentence highlighting insights about potential mechanisms in this paragraph (lines 685–689).

Reviewer #3

(Public repository details (Required)): 16S sequencing data

We have now uploaded these data in the NCBI Sequence read archive and the accession numbers are now added in the manuscript (line 712).

The authors tested the effect of native vs non-native *Methylobacterium* inoculation on three different rice varieties in a field trial. They measured a large number of different traits and found that the results varied between variety and strain inoculation, indicating that results are not generalizable across *Methylobacterium* strains and rice landraces. They then go on to explore different mechanisms that might explain these variations but encounter mostly negative results. Due to the setup of the experimental field design, the interpretation of the results is limited. Apart from the "other" strain, *Methylobacterium* strains have been tested on a single landrace. However, the authors are upfront about the limitations and mostly careful with their interpretation. The manuscript is largely well and clear written. There are a few clarifications (especially regarding figure design) that would improve the manuscript and help in the interpretation of the results.

Thank you for your encouraging comments.

Comments:

1. Line 13: "dominate" might be a strong word, I would suggest rather "abundantly found" or similar **We have now corrected as suggested.**

2. Line 22: "than the bacterial growth medium on its own" **We have now corrected as suggested.**

3. Line 25: remove "clear"

We have removed as suggested.

4. Line 42: "inoculation with some bacteria..." as you do not show that these bacteria directly increase rice grain production and especially the non-native strain itself reduces plant health.

We have now corrected as suggested.

5. Line 142-143: What OD can the different strains reach within these 48 hrs? Are they stationary at that point and are similar in yield?

Yes, they are in stationary phase and have similar OD (line 143).

6. Which landrace was grown on the plot beforehand? Was it one of the tested landraces?

Yes, we chose this particular field since it is often used to grow Chakhao and Phouren-mubi.

7. Line 180: 45 days after transplantation -> what is the growth stage at that time point?

There is no defined stage at this point of vegetative growth, but we have now explained that the plants had 6-8 green leaves at this point (line 181).

8. Line 185: I assume you added an antifungal compound to the plates?

No, we did not add antifungals.

9. L189-199: Have you tested whether all bacteria are reliably dislodged from the leaf cuttings?

This is a good point but it is actually nearly impossible to conclusively test this. If we were to sequence the leaf tissue, we would not be able to account for endophytic bacteria living inside the plant tissue. However, we did do the next best thing, by washing the processed leaf with sterile water and plating on bacterial growth media. Since we did not observe any growth, we concluded that most of the epiphytic bacteria were successfully dislodged while processing.

10. L215: How deep did you sequence?

Thank you for pointing out this omission. We obtained minimum 32000 reads per sample after filtering steps. We have now added this in the methods section (line 224).

11. L217-231: From your materials and methods section, I gather that you did the analysis at 97% OTU level rather than at the level of ASV. Could you please specify?

We used ASV analysis using the DADA2 workflow, which uses 100% sequence identity. We have now mentioned this in lines 220-222.

What is the reasoning behind clustering *Methylobacterium* spp at 97% sequence identity to check whether your inoculated strains can be detected on rice plants but using 100% sequence identity when looking at pathogen presence? I would be better to use ASVs to make an assumption about the presence of your inoculated strains as 97% sequence identity is generally only species level.

Sorry about this confusion. We actually used 97-100% sequence identity for both *Methylobacterium* strains as well as for pathogens. However, it so happened that all the pathogens were 100% matched with known bacterial pathogen sequences (this is now more clearly conveyed in line 234).

I am also a bit puzzled by your use of Chi-square tests in e.g. in Fig 5. Since you are working with compositional data from sequencing, other tests would be more appropriate. I would encourage you to use e.g. edgeR or DESeq2 for checking for differences in an OTU between treatments.

We agree that the more sophisticated tools suggested above may be better. However, since we were only interested in two or three focal taxa, it was much simpler to conduct chi-square tests to analyse changes in the relative proportions off the focal taxa across treatments.

12. You write that you use GLMs for data analysis. What link function did you use? How did you control for multiple testing of the large number of traits you looked at?

We used normal error distribution for our GLMs; we have now mentioned in this in line 290. We used Tukey's HSD to account for multiple comparisons (lines 290-292).

13. Fig. 3A: I suggest using continuous y-axes for all landraces to reduce confusion, especially since there does not seem to be a need for split y-axes. Also, since this shows difference in growth between

T3 and T1 and growth rate does not seem to be constant (based on Fig S3), I would suggest to show absolute difference in size rather than growth rate.

We use the split axes to ensure that the data points are visible. Without the split axes (with the y-axis beginning at 0), all the data get scrunched up at the top and are not easily visible. Each landrace has a different baseline height, which makes comparison of bacterial effects across landraces difficult. Hence, we chose to show the growth rate across land races, which is easily comparable. The absolute heights are shown in Fig S4, for interested readers.

14. Fig. 3B: Why are there less samples than in A? Did you not determine grain yield for all plants? What is the difference between Fig S5A and Fig 3B? For me it looks like the same data is depicted but significance asterisks differ between the two.

Yes, there are less samples in the grain yield data because some of the plants were infested with fungus while transporting back in lab and we excluded them from the yield analysis. We have now mentioned this (lines 192-194).

The two figures are indeed identical, except that in figure S5 we also included the impact of statistically influential data points. There was however, a mistake in placing asterisks in Fig S5 and we have now corrected this. Thank you for spotting these errors.

15. L 356: Based on Fig S4 only total plant height tends to be decreased relative to untreated. Relative to the broth control most measurements are either similar or higher for the "own" treatment and only the "other" treatment might be lower. Thus, the conclusion that *Methylobacterium* reduce vegetative growth across land races seems not in line with the data.

Thank you for spotting this error. We have now corrected the sentence to say "little impact" (line 361).

16. For your graphs, you generally indicate significant differences relative to the broth control only. As the broth control is not the left most treatment and not labelled as "the" control this is a bit confusing because one would thus assume that e.g. own vs untreated is not significantly different when there is no significant asterisk indicated. Can you please comment?

We understand the concern. In the main text, we had elected to highlight comparisons with the broth control rather than the untreated control, because the latter seemed to be a better choice as a baseline expectation and we thought it would be confusing for readers to be shown all comparisons in the figures.

We had, however, compared treated plants to untreated plants, and have now added the results of this statistical analysis to Table S4, for interested readers.

17. Fig. 4: I would recommend to show the differences as (log)-fold-changes rather than in absolute values. It is difficult to compare for example a difference of 5 in flag leaf length to a difference of 300 in number of filled grains since the scales between these differ strongly. These graphs do not give easy access to the effect sizes. Also, for "own" you compare two different bacterial treatments, whereas for "other" it is the same. It would be helpful to have this information in the legends. Furthermore, Table S4 and not Table S3B should be referenced in the text along Fig. 4.

We understand the concern, and we struggled with this visualization ourselves. However, we find it very difficult to interpret log fold changes rather than absolute values of traits, and hence we choose to display the latter and used a split axis that allows readers to visualize both small and large differences. Effect sizes are also more easily seen in the boxplots showing median values for each trait (Fig 3 and Figs S4 and S5).

For both "own" and "other", we actually compare one bacterial treatment with the broth only treatment.

Thank you for pointing out the correct citation should be Table S4; we have now fixed this.

18. L392: "can be beneficial"

We have corrected as suggested.

19. The recovered *Methylobacterium* strains that you had from plating, did you only pick different morphologies or how did you quantify these? From Fig S6 it looks like you have for example many pink colonies on plant 3 of untreated Phou-ngang or in the own "Chakhao" but Table S5 mentions only few total isolates for the different ones. Also, are the "own" and "other" here based on 100% sequence identity of the 16S?

Yes, we picked a few distinct pink colonies from the plates to confirm the identity. We have now specifically mentioned this in the legends to Table S5 and Fig S6.

Yes, the identification was based on 100% sequence match with our inoculated strains. We have now mentioned this in the legend to Table S5.

20. L439: all your effects could also be explained by effects on other microbes present in the community and not by direct interactions with the host. Furthermore, they could for example also be caused by differences e.g. in pH or other parameters of your growth medium that is applied together with your strains. Have you checked for differences between the strains? Or what happens when you wash bacteria before you inoculate them?

We see the concern, and have rephrased to say “immediate impacts on the host” instead of “interactions with the host” (line 444).

Unfortunately, we did not check the pH of growth media before inoculation, or the washing experiment suggested here.

21. Fig. 5: Have you tested whether rel. pathogen abundance differs between own and other? For me, it looks like it is similar between the two, thus I would not specifically mention that pathogen proportion was lowest in "other" treatment. For panel C please label y axis to make clear that we are only looking at the relative abundance of the total M.

Thank you for pointing out this error. We have now deleted the mention of lowest pathogen abundance in “other” treated plants (line 459).

We have now specified “total M” in the legend for Fig 5C.

22. For the microbiome analysis when you specifically look at the Methylobacterium proportion, you group them into "own", "other" and "environmental". How do you treat then an "own" of e.g. Phouren-mubi on Phou-ngang? Would this be counted as "own" or not? I am wondering whether on Phou-ngang the inoculated strain itself is just less fit and the environmental could be a different strain.

“own” is counted with respect to each landrace, and hence, for Phou-ngang, the “own” of Phouren-mubi would be counted as “environmental”. It is quite possible that the Phou-ngang “own” strain is generally less fit.

23. L 465: I would claim that the relative abundance of "own" in Phouren-mubi does not look really lower than in Chakhao. Did you do an analysis?

Here, we are stating that the “own” strains were not abundant, without making a comparison with Chakhao (line 469).

24. L521: Did you check specifically whether flagella are absent in your Methylobacteria? Because there are Methylobacteria that contain flagella.

The point is well taken and we have rephrased the wording in the manuscript (line 526). Incidentally, we had done some SEM imaging of our strains and we did not observe flagella.

25. In the introduction you go quite a bit into detail about cytokinin production by Methylobacterium spp but this aspect is then neither tested nor further discussed as a possible mechanism. Why did you test auxin and not cytokinin production?

Auxin is one of the important hormones that plant-associated microbes secrete to breach the cell wall and help in colonization (Vorholt 2012, Nat Rev Microbiol). Hence, we tested for auxin production. It would absolutely be useful to test for cytokinins as well, but we were unable to find an assay method that was logistically possible for us to conduct. We have now suggested that future work should also test for cytokinins (line 593).

26. Table S1: There seems to be a mistake in the amount of succinate supplied per plant

We have corrected this, thank you for catching it.

June 24, 2022

Prof. Deepa Agashe
National Centre for Biological Sciences
Bangalore
India

Re: Spectrum00810-22R1 (Impact of phyllosphere *Methylobacterium* on host rice landraces)

Dear Prof. Deepa Agashe:

Sorry for the delay. I have read over your responses and edits and feel you have reasonably addressed the concerns raised by the reviewers.

Your manuscript has been accepted, and I am forwarding it to the ASM Journals Department for publication. You will be notified when your proofs are ready to be viewed.

Sincerely,

Erik Hom
Editor, Microbiology Spectrum

Journals Department
Supplemental Dataset: Accept
Supplemental Material: Accept